# Differential Gene Expression Analysis of Wheat Breeding Lines Reveal Molecular Insights in Yellow Rust Resistance under Field Conditions

**Sandeep Kumar Kushwaha [1,2,\*], Ramesh R. Vetukuri [1], Firuz Odilbekov [1], Nidhi Pareek [3], Tina Henriksson [4] and Aakash Chawade [1]**

[1] Department of Plant Breeding, Swedish University of Agricultural Sciences, 230 53 Alnarp, Sweden; ramesh.vetukuri@slu.se (R.R.V.); firuz.odilbekov@slu.se (F.O.); aakash.chawade@slu.se (A.C.)

[2] National Institute of Animal Biotechnology, Hyderabad 500032, India

[3] Department of Microbiology, School of Life Sciences, Central University of Rajasthan, Bandarsindri, Kishangarh, Ajmer, Rajasthan 305801, India; nidhipareek@curaj.ac.in

[4] Lantmännen Lantbruk, 268 31 Svalöv, Sweden; tina.henriksson@lantmannen.com

**\*** Correspondence: sandeep@niab.org.in; Tel.: +40-2312-0150; Fax: +40-2312-0130

**Abstract:** The evolution of pathogens in the changing climate raises new challenges for wheat production. Yellow rust is one of the major wheat diseases worldwide, leading to an increased use of fungicides to prevent significant yield losses. The enhancement of the resistance potential of wheat cultivars is a necessary and environmentally friendly solution for sustainable wheat production. In this study, we aimed to identify the differentially expressed genes induced upon yellow rust infection in the field. Reference and de novo based transcriptome analysis was performed among the resistant and susceptible lines of a bi-parental population to study the global transcriptome changes in contrasting wheat genotypes. Based on the analysis, the de novo transcriptome analysis approach was found to be more supportive for field studies. Expression profiles, gene ontology, KEGG pathway analysis and enrichment studies indicated the relation between differentially expressed genes of wheat and yellow rust infection. The h0igh expression of genes related to non-race specific resistance along with pathogen-specific resistance might be a reason for the better resistance ability of a resistant wheat genotype in the field. The targeted metagenomic analysis of wheat samples revealed that *Puccinia striiformis tritici* was the most dominant pathogen along with other pathogens on the collected leaf material and validating the disease scoring carried out in the field and transcriptomics analyses.

**Keywords:** wheat; yellow rust; disease resistance; transcriptomics; differential gene expression

## 1. Introduction

Yellow (stripe) rust (YR) in wheat (*Triticum aestivum* L.) is caused by the biotrophic pathogen *Puccinia striiformis f.* sp. *tritici* (Pst) and is distinguished by the yellow pustules that occur on leaves. It is a major disease of wheat worldwide and if left uncontrolled, may cause 100% yield losses in highly susceptible wheat cultivars [1]. The disease is airborne and spread through urediniospores which can disperse over long distances by wind [2]. New evolving races are regularly identified in north-western Europe and have become a major concern in recent years [3]. Genetic resistance to yellow rust in wheat at the seedling stage is mainly through the major resistance (R) genes while at the adult-plant stage, the genetic resistance can be either from R genes or adult-plant resistance genes (APR) or both. R genes recognize and neutralize specific pathogen effectors in a gene-for-gene

interaction resulting in effector-triggered immunity (ETI) in plants leading to complete disease resistance [4]. R genes are easy to incorporate in breeding programs, and they can provide high-level resistance to YR. However, they are not durable due to the evolution of new virulent races [5]. Several historic epidemics were also observed in the past due to the breakdown of resistance to YR [6]. On the contrary, adult-plant stage resistance is due to the additive effects of several genes leading to partial quantitative resistance. Adult plant quantitative resistance is more durable due to its complex genetic nature and varieties with high and durable resistance can be developed by incorporating both R and APR genes [7]. Resistance genes for yellow rust are designated as *Yr*, and several of these are responsive mainly at the seedling stage while some are effective at the adult plant stage [7]. To date, 82 *Yr* genes have been identified, and around 25 of them are related to APR or high temperature adult plant resistance, whereas the rest provide all stage resistance (ASR) [8]. Several of the ASR genes (*Yr5*, *Yr15*, *Yr53*, *Yr61*, *Yr65* and *Yr69*) are still effective and can be used in breeding for YR resistance [9]. Although these *Yr* genes have been genetically mapped to wheat chromosomes, only a small number of them have been isolated to date [9]. More than 140 QTL (Quantitative trait locus) have been identified with partial quantitative resistance to yellow rust and are distributed throughout the wheat genome [10]. A daunting task is to identify the candidate resistance genes within these QTL regions.

The transcriptome profiling and quantification of the differential expression of genes and proteins play a key role in the elucidation of regulatory pathways and gene-networks through wide transcripts coverage, high sensitivity, allele-specific differential expression and novel transcript identification [10–14]. Transcriptome analysis has facilitated the fine mapping of the *Yr* genes and in understanding the underlying regulatory mechanisms in several studies. Coram et al. (2010) identified 28 transcripts commonly induced upon race-specific resistance conferred by eight different *Yr* genes [15]. Hulbert et al. (2007) identified 59 putative rust-induced transcripts expressed in the flag leaves of the spring wheat Thatcher-Lr34/Yr18 isoline [16]. Chen et al. (2013) identified 102 and 113 rust responsive wheat genes associated with *Yr5* and *Yr39* genes, respectively [17]. Comparative transcriptomics revealed distinct differences in the responsive genes upon yellow rust and powdery mildew infection [18,19]. Transcriptomics studies have a high potential for innovative and exploratory studies towards novel insights into molecular mechanisms. However, a large number of genes identified in large scale omics studies impose a hurdle to identify candidate genes for functional validation. A map-based cloning approach has been successful at cloning key *Yr* genes, and induced mutagenesis identified Lr34/Yr18 to be an ABC transporter [20]. Map-based cloning in 4500 F2 plants followed by chemical mutagenesis and the screening of 1536 M2 lines led to the identification of a gene, *Yr36*, with a kinase and START domain [21]. *Yr10* was identified as a CC-NBS-LRR domain-containing protein from a mapping population of 874 BC2F3 individuals [22]. In this work, we evaluated the possibility to identify candidate genes for *YR* resistance at the adult plant stage in a winter wheat bi-parental population through transcriptomics approach and also performed the metagenomics analysis to identify major wheat pathogens in the field trial.

## 2. Materials and Methods

### 2.1. Field Experiments and Sampling

A winter wheat bi-parental population of 109 lines obtained from the cross (*Nimbus/3/SW, 2081221/2/SW2-7/Kranich*) was sown in the field in Svalöv, Sweden, in the autumn of 2013. Scoring for yellow rust resistance was done at the booting stage (Zadoks stage 41–49) in the spring of 2014 with a scale of 1 (high resistance) to 9 (highly susceptible). On the basis of scoring, leaf materials from resistant and susceptible lines were collected from penultimate leaves pooled from three plants from each breeding line of the segregating population at the booting stage (Zadoks stage 41–49). Leaves were flash-frozen in liquid nitrogen and stored at −80 °C until further processing. Total RNA was extracted with the RNeasy Plant Mini kit (Qiagen, Hilden, Germany) including DNase treatment (RNase-free DNase set, Qiagen Inc. Santa Clarita, CA, USA).

### 2.2. RNAseq Library Preparation Sequencing and Quality Control

The concentration and quality of the RNA were estimated in ExperionTM Automated Electrophoresis System (Bio-Rad Laboratories, Hercules, CA, USA) and RIN values above seven were used to construct separate cDNA libraries. Paired-end sequencing was performed for 25 wheat lines on Illumina HiSeq 2500 instrument at the SciLifeLab (Stockholm, Sweden) with 300 base-pairs (bps) average fragment length. The FastQC program was used to analyse the quality of the raw sequencing reads [23]. Adapter sequences and low-quality reads were removed using trimmomatic [24].

### 2.3. De Novo and Reference Transcriptome Assembly and Validation

All the clean reads were mapped to the 13 known wheat pathogens (*Zymoseptoria tritici (Desmazières) Quaedvlieg & Crous, Pyrenophora tritici-repentis (Diedicke) Drechsler, Bipolaris sorokiniana (Ito & Kuribayashi) Dastur, Puccinia graminis Persoon, Puccinia striiformis Westendorp, Puccinia triticina Eriksson, Phaeosphaeria nodorum (Berkeley) Quaedvlieg, Verkley & Crous, Pyrenophora tritici (Diedicke) Drechsler, Fusarium culmorum (W.G.Smith) Saccardo, Fusarium graminearum Schwabe, Fusarium oxysporum Schlechtendal, Fusarium verticil (Saccardo) Nirenberg* and *Magnaporthe oryzae Cavar*) and un-aligned reads derived from individual lines were pooled to construct reference and de novo assemblies. De novo assembly was generated by using the Trinity assembler (version 2.5.1) [25]. Trinity pipeline was followed for the detection of differentially expressed genes. To facilitate a more in-depth comparison of the two different genotypes, reference-based transcriptome assembly was also generated through the alignment of all clean reads from each line to the reference wheat genome by using HiSat2 and Stingtie software [26]. The BUSCO software (version 2.4) was used to evaluate the quality of the two transcriptome assemblies [27]. Generated transcripts were clustered on 100% sequence identity through CD-HIT software [28]. Quality check of samples were also performed through   expression abundance estimation of lines, PCA and MDS plot and YR field scoring of lines were also compared. High scoring (susceptible) and low scoring (resistant) lines were used as replicate for differential gene expression analysis.

### 2.4. Identification of Differentially Expressed Genes (DEGs), Annotation and Gene Ontology (GO) Analysis

The de novo and reference assembled transcriptome was then used as a reference to map the individual reads using the Bowtie2 program [29]. The transcript abundance, raw, transcript per million (TPM) and fragments per kilobase (kb) of the transcript sequence per million mapped reads (FPKM) was measured by using RSEM version 1.1.1131 for each sequenced line [30]. The DESeq2 package was employed to identify of differentially expressed genes (DEGs) from raw read counts at a false discovery rate (FDR) of 0.05 [31]. The heatmap of DEGs was generated through Pheatmap R package by using euclidean distance and hierarchical clustering algorithm [32]. The annotation of the DEGs was performed using the BLAST search program. Initially, BLASTx was performed with an e-value threshold of 1e-10 against wheat coding sequences (CDS) and Uniprot database. The Gene Ontology (GO) classification of DEGs in the genotypes were generated using the WEGO program [33]. The GO enrichment analysis was performed through AgriGO webserver by using hypergeometric statistical test methods and Yekutieli multiple test correction at a *p*-value of 0.05 [34], and visualization was done through REVIGO [35]. A KEGG pathway analysis was performed with the KOBAS program [36].

### 2.5. Metagenomic Analysis of Selected Wheat Pathogens

Targeted metagenomics analysis was performed to explore wheat-associated pathogen species in the samples. The trimmed RNA libraries were mapped to 13 selected wheat pathogens with Bowtie2 mapping software. The statistical significance of mapped reads were evaluated for pathogens with respect to resistant and susceptible groups.

## 3. Results

### 3.1. RNAseq Data Analyses

The resistant and susceptible lines were sequenced and analysed by the reference and de novo transcriptome assembly. A total number of 307 million read pairs (2x150 bps) were generated from the 20 lines. After quality control, 207 million high-quality reads of average length 126 bps were used in the construction of transcriptome assemblies. Library size, gene expression in lines, principal component analysis (PCA) plot and multidimensional scaling (MDS) plots of the samples are given in Figure 1. All the high quality wheat reads were pooled for the construction of reference and de novo assemblies and generated assemblies were clustered at 100% sequence identity. After clustering, 219,435 (N50 of 1835 bps) and 511,926 (N50 of 1420 bps) contigs were found for reference and de novo assemblies respectively. The longest contig sizes were 22,462 and 15,387 bps, respectively (Table 1). In BUSCO analysis, reference and de novo transcriptome assemblies achieved the 94.7 and 78.5% completeness, respectively. The percentage of partially complete BUSCO ranged from 99.1 to 94.0%, while the percentage of missing BUSCO was 0.9 and 6.0%, respectively (Table 2). The wheat lines, disease scoring, read count per line, high quality read, and mapping percentage are given in Supplementary Table S1.

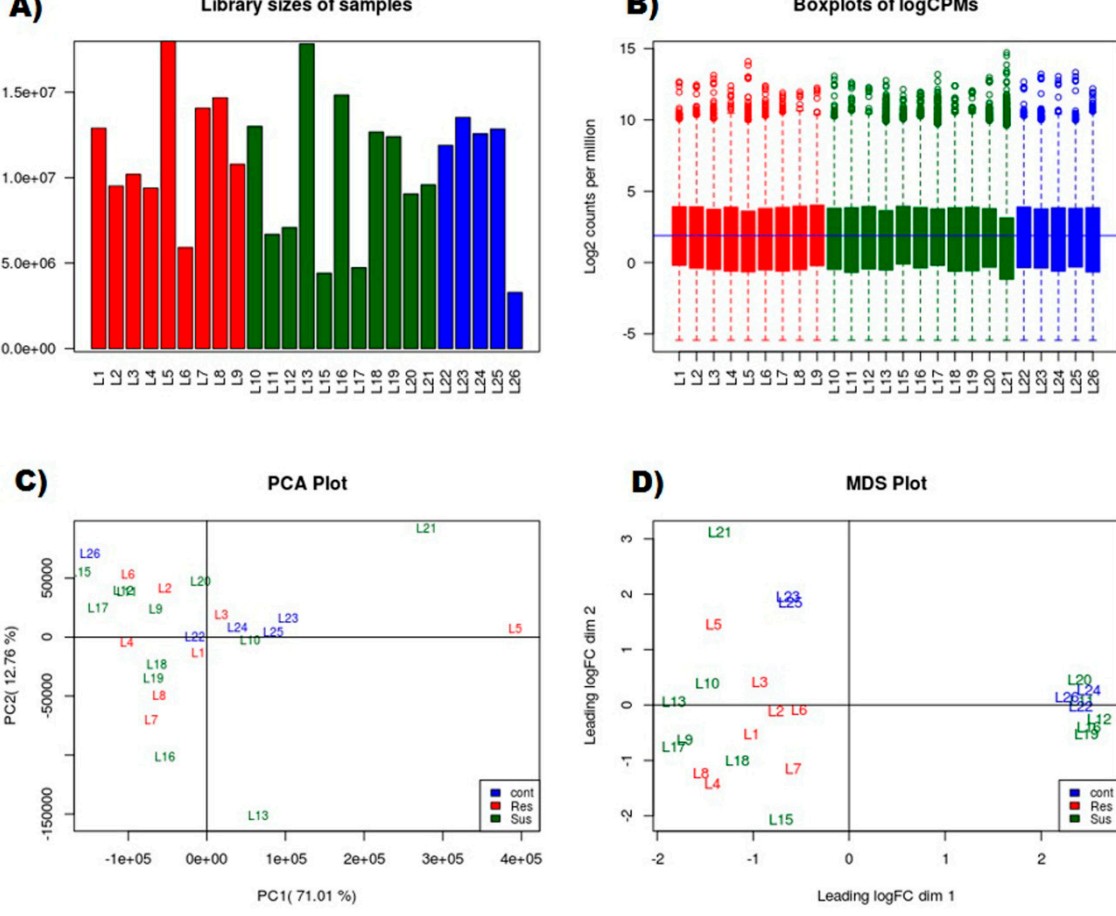

**Figure 1.** Quality evaluation of the samples: (**A**) library size of the sequenced wheat lines; (**B**) boxplot of logCPM expression values across the samples; (**C**) principal component analysis (PCA); and (**D**) multidimensional scaling (MDS) plot of the Trimmed Means of M-values (TMM) normalized expression values of samples. Resistance, susceptible and control lines are represented through red, green and blue colours, respectively. Control lines (cont): blue, resistant lines (Res): red, susceptible lines (Sus): green.

**Table 1.** Assembly metric of the reference and de novo transcriptome assemblies.

| Transcriptome | Contigs | N50 | Avg. Length | Assembly Size | Longest Contig size | Contig Size >10 KB | Contig Size >1 KB |
|---|---|---|---|---|---|---|---|
| Reference | 219,435 | 1835 | 1477 | 324,000,551 | 22,462 | 58 | 140,664 |
| De novo | 511,926 | 1420 | 953 | 488,058,762 | 15,387 | 30 | 175,517 |

**Table 2.** Results of the BUSCO analysis for the transcriptome assemblies' validation.

| BUSCO Description | Reference | De Novo |
|---|---|---|
| Complete BUSCOs (S+D) | 1302 (94.7%) | 1079 (78.5%) |
| Complete and single-copy BUSCOs (S) | 69 (5.0%) | 229 (16.7%) |
| Complete and duplicated BUSCOs (D) | 1233 (D: 89.7%) | 850 (D: 61.8%) |
| Fragmented BUSCOs (F) | 61 (4.4%) | 213 (15.5%) |
| Missing BUSCOs (M) | 12 (0.9%) | 83 (6.0%) |

*3.2. DEGs Identified by De Novo and Reference-Based Methods*

To quantify the transcriptomic variations in a sequenced line, we aligned the clean reads from each sample against the wheat reference and de novo transcriptome assemblies by using Bowtie2 with default parameters and RSEM software was used to quantify the transcript abundance to compare the expression level within and between different samples. The DEGs identified in the reference and de novo assembled transcriptome by DESeq2 software in pairwise comparisons between the resistance and susceptible lines were 141 and 8680, respectively, with a false discovery rate (FDR) <0.05 (Table S2). To find wheat genes in the reference and de novo based analysis, identified DEGs were searched against wheat CDS sequences through BLAST similarity search, including up and downregulated transcripts (Figure 2). To determine the sample relations, differential expression data from the DESeq2 program were used to generate heat maps. Resistant and susceptible lines were grouped together in a different order with respect to reference and de novo transcriptome. However, most of the susceptible and resistant lines were grouped in the same clusters in both transcriptomes (Figure 3A and 3B). The expression heatmap of differentially expressed genes was clearly indicating a fewer number of upregulated genes in resistant lines than susceptible lines in their respective assemblies.

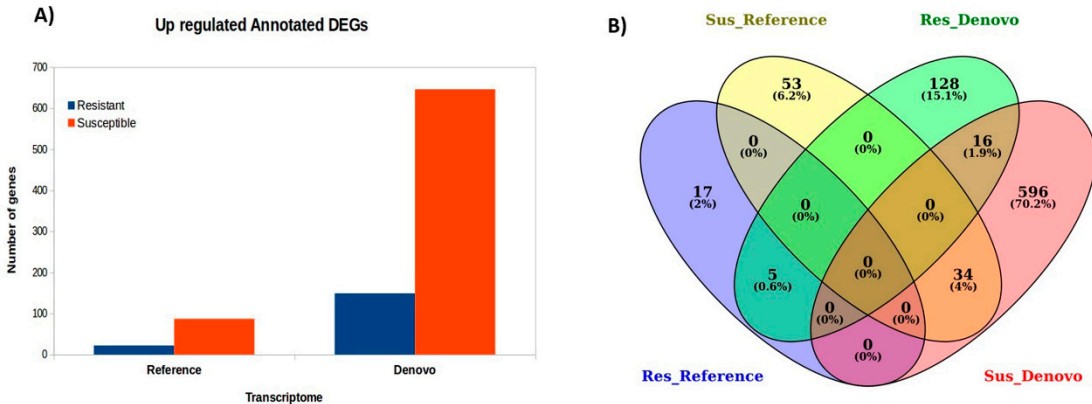

**Figure 2.** Summary of differential gene expression and comparison: (**A**) the number of differentially expressed transcripts identified using reference-based assembly and de novo based assembly; (**B**) Venn diagrams of the number of differentially expressed transcripts for pairwise comparisons at a false discovery rate (FDR) of <0.05.

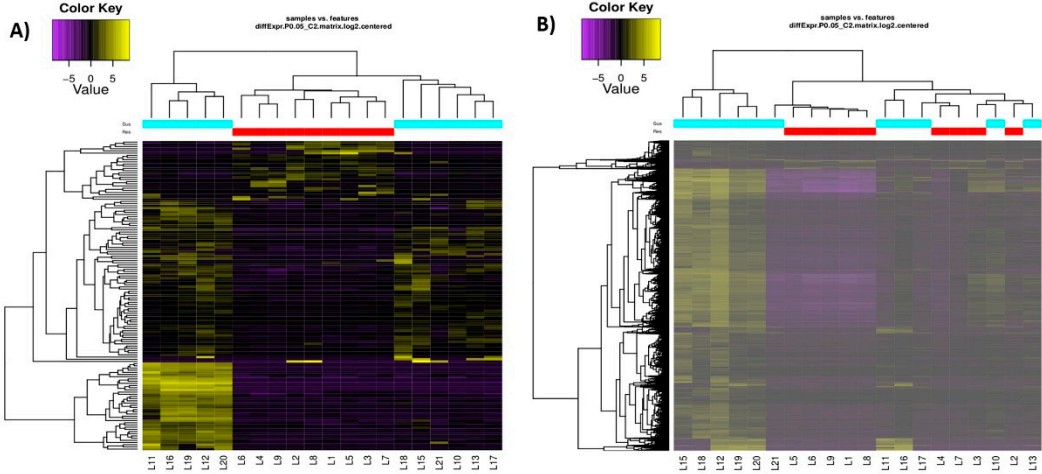

**Figure 3.** Heat maps of the expression pattern of differentially expressed genes by using (**A**) 141 reference and (**B**) 8680 de novo assemblies. Red and cyan represent the resistant and susceptible lines, respectively.

### 3.3. Differentially Expressed Genes Annotation and GO Enrichment

A BLASTX similarity search was performed for all the identified DEGs against the Uniprot database. Approximately 53% of the DEGs had BLASTX hits. The majority of the top BLASTX hit species in the reference-based DEGs belonged to *Arabidopsis thaliana*, *Oryza sativa subsp. Japonica*, and were followed by other cereal crops. In the de novo based analysis, DEGs were identified in approximately 17% of plant species, mainly: *Arabidopsis thaliana*, *Oryza sativa janonica*, *Zea mays*, *Triticum aestivium,* and *Nicotiana benthamiana;* 35% *Schizosaccharomyces prombe* and *cerevisiae;* 20% of fungal species, such as *Puccinia gramine*, *Ustilago maydis*, *Cryptococcus* neoformans and *neoformans* sp., *Dictyostelium discoideum*, *Neurospora crassa*, *Candida glabrata* and *albicans* sp., *Neosartorya fischeri*, *Pneumocystis carinii*, *Solanum bulbocastanum*, *Trichosporon cutaneum*, *Agaricus bisporus*, *Aspergillus niger*, *flavus*, *terreus* and *oryzae*, *Chaetomium thermophilum* and *globosum* sp., *Pseudomonas aeruginosa*, *Eremothecium gossypii*, *Debaryomyces hansenii*, *Phanerochaete chrysosporium*, *Uromyces fabae*, *Laccaria bicolor*, *Kluyveromyces* sp., *Bacillus subtilis*, *Yarrowia lipolytica*, *Emericella nidulans*. All the reference and de novo assembled transcripts were annotated against wheat genome and wheat annotated differentially expressed transcripts were considered for further downstream analysis. Wheat annotation for transcripts can be found in Table S3 and Table S4.

The Gene Ontology (GO) classification of DEGs' was represented in three main GO categories, i.e., the cellular component, molecular function and biological process in a histogram (Figure 4). In the GO analysis, the cellular component, cell and cell part (80%), organelle (55%) and membrane (46%) were highly represented compared to other components. Binding (64%) and catalytic activity (61%) were most represented among the molecular functions. Cellular process (65%), metabolic process (56%) and response to stimulus (36%) were the most dominant subcategories of biological processing. Cellular biosynthetic process, metabolic process, i.e., macromolecule, organic substance, nitrogen compound, aromatic compound metabolic process, and organic cyclic compound metabolic process were highly enriched in GO analysis. The significantly enriched GO term of the biological process can be further subcategorized into response to stress, response to stimulus, signal, methylation, multicellular organismal processing, multi-organism process and reproductive process which can be seen clearly in Figure 5A, generated by REVIGO. In figure 5A, the X and Y coordinates were based on the multidimensional scaling of a matrix of the GO terms' semantic similarities whereas bubbles closeness on the plot were reflecting their closeness in the GO graph structure. It is clear that response to stress, stimulus, signalling and molecular localization-related terms were among the enriched GO terms. The KEGG database in KOBAS webserver was used to explore the networks of enriched pathways and gene products (Table S5). An enriched pathway among differentially expressed genes was provided as a scatter diagram with a degree of enrichment by the

rich factor, p-adjust, and the number of genes enriched in a pathway (Figure 5B). The richness factor represents the ratio of the quantity of genes belonging to the pathway among differentially expressed genes to the total number of genes belonging to the pathway among all annotated genes. The lower the *p*-value, the higher the significance of metabolic pathways, whereas the size of bubbles represents the number of enriched genes. Metabolic pathways, the biosynthesis of secondary metabolites, peroxisome, glycerolipid metabolism, the mRNA surveillance pathway, plant–pathogen interaction, cutin, suberine and wax biosynthesis, glycerophospholipid metabolism and ubiquitin-mediated proteolysis have appeared as highly enriched pathways. To explore global transcriptome expression, RNAseq data of wheat resistant and susceptible lines were mapped on wheat genome. In general, genes in susceptible lines have a higher average read depth per million base pairs than resistant lines and an approximately similar trend has also been found at the genomic location of differentially expressed genes (Figure 6). However, higher read depth in resistant lines for differentially expressed genes was also found on wheat genome 1A, 2A, 3A, 3D and 5A, compared to susceptible lines which might be helpful to understand the molecular mechanism of resistance among the resistance lines.

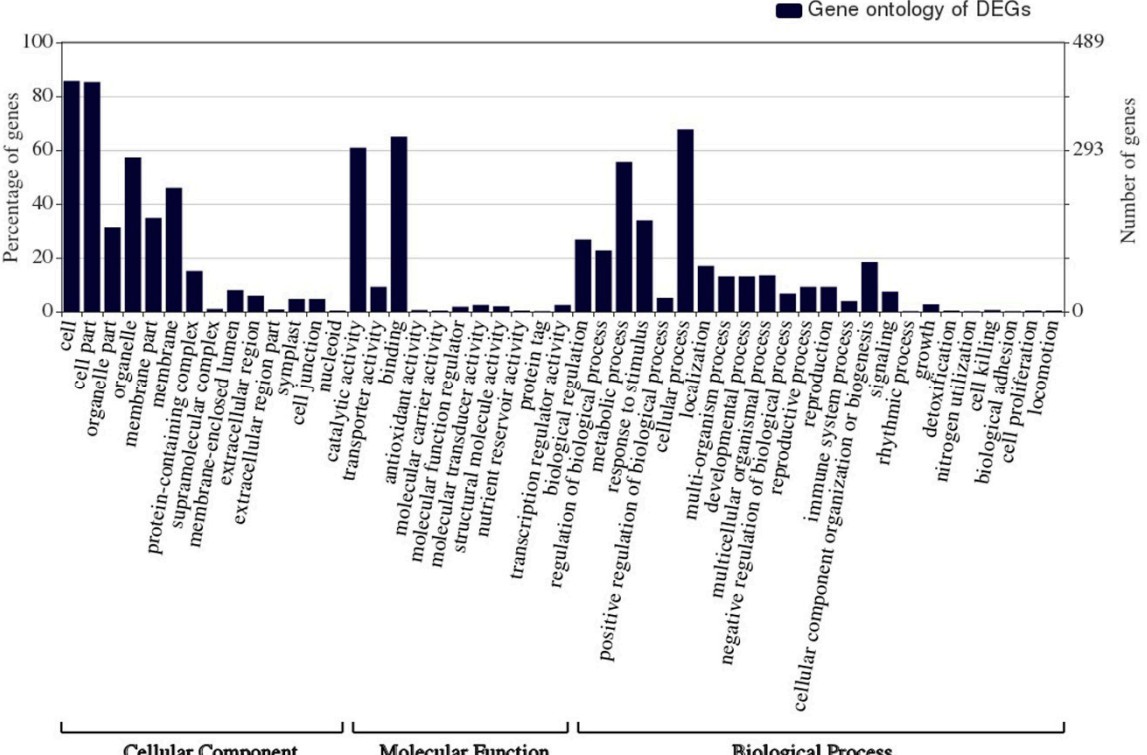

**Figure 4.** Gene ontology classifications of the differentially expressed genes by the WEGO tool.

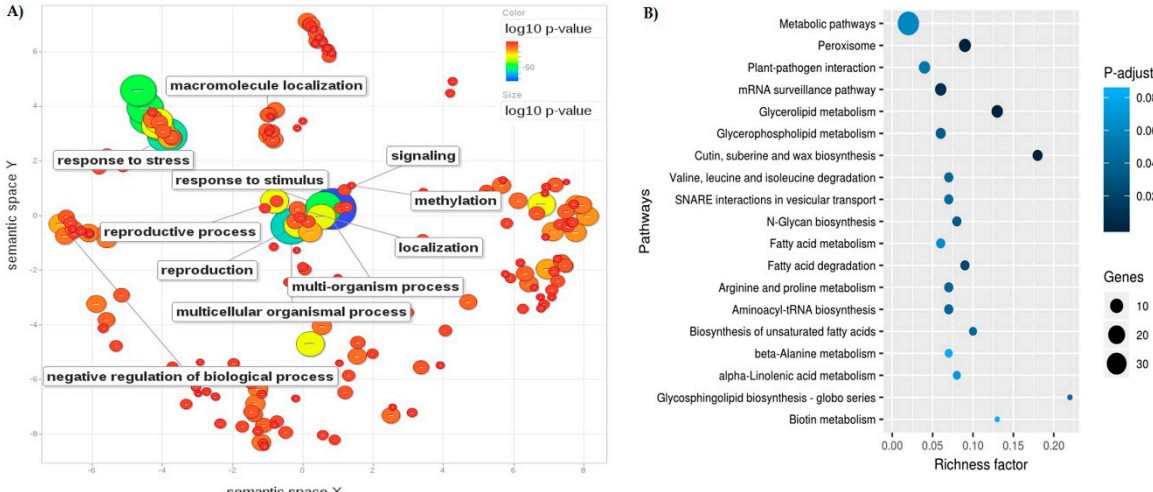

**Figure 5.** Gene ontology and pathways: (**A**) description of enriched GO terms associated with differentially expressed genes through a scatterplot in a two-dimensional space derived by applying multidimensional scaling to a matrix of the GO terms' semantic similarities. The bubble colour indicates significance (−log10 *p*-value) and size indicates the frequency of the GO terms in the underlying gene ontology annotation database, such as a larger circle in blue which represents the most significant enriched term. Red colour represent higher p-values (**B**) The KEGG enrichment of differentially expressed genes as a scatter diagram with a degree of enrichment by the richness factor, *p*-adjust, and the number of genes enriched in a pathway. The number of enriched DEGs in the pathway is indicated by the circle area, and the circle colour represents the ranges of the *p*-adjust.

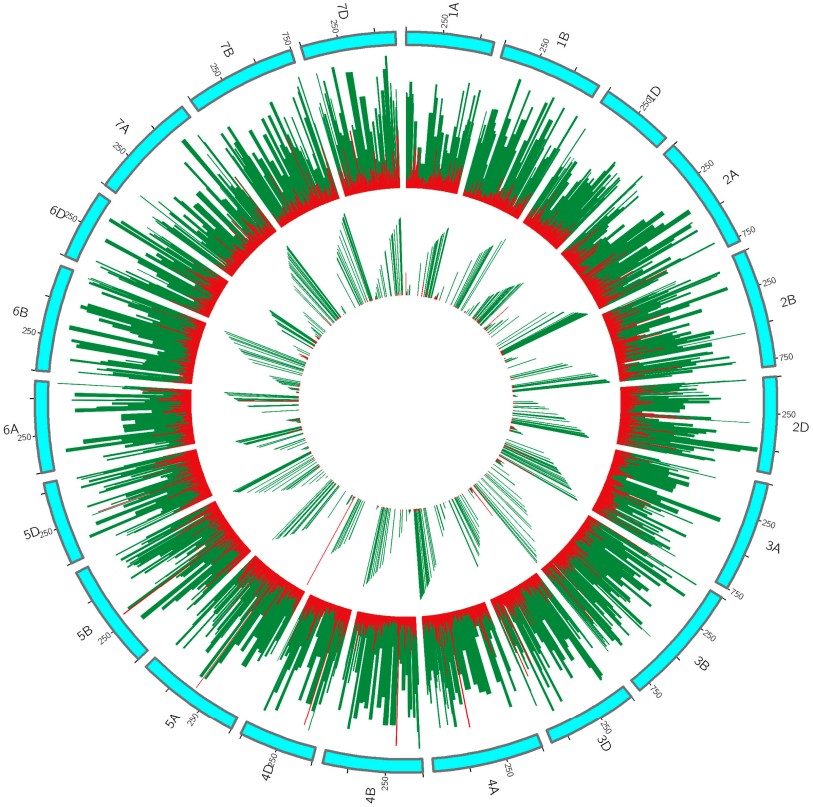

**Figure 6.** Circos visualization of the RNAseq data at the wheat genome-wide level. From the outer to inner ring; karyotype of the wheat genome; average read depth per million base pair of wheat genome mapping of the resistant (red) and susceptible (green) line; the genomic location of differentially expressed genes for the resistant (red) and susceptible (green) line. Height of bars in outer and inner ring shows the average read depth per million base pair with respect to the genomic location.

To explore the field pathogenic factor association with the gene expression of wheat lines, targeted metagenomic analysis was performed by read mapping of resistant and susceptible line samples to the 13 known wheat pathogens. The highest number of reads was mapped to *P. striiformis* genome in the susceptible group (Figure 7). Significant differences in the read counts in the susceptible and resistant groups were also found in between those of *P. graminis* and *P. triticiana*, however, none of the wheat samples had more than 0.5% of total reads belonging to either of these two pathogens. RNA sequences from the pathogens *B. sorokiniana*, *F. culmorum* and *Z. tritici* were also identified with read counts of at least 1% of total reads in at least one sample. However, the read counts from these pathogens were not significantly different between the resistant and susceptible groups.

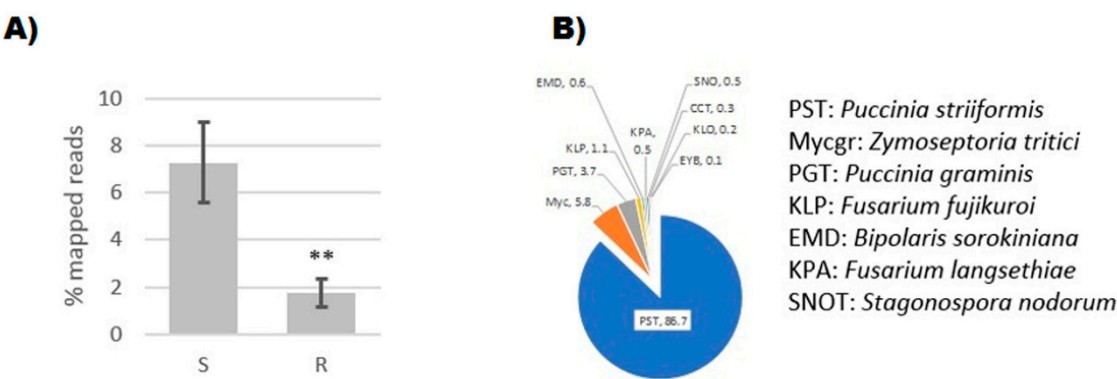

**Figure 7.** Metagenomic analysis of the selected wheat pathogens: (**A**) the percentage of pathogen reads that are mapped to each species (*** $p < 0.001$; ** $p < 0.01$); (**B**) the mapping percentage of reads for the pathogenic fungal genome.

## 4. Discussion

In this study, resistant, susceptible and control wheat lines were sequenced. Control lines, common filed trial lines were used to check disease pressure in field whereas resistant and susceptible lines were selected on the basis of field scoring. To compare the transcriptomic variations among wheat lines exposed to yellow rust disease in the field, leaf tissue samples were collected and used for the RNA-seq analysis. On average, approximately 69% of reads were mapped to the wheat reference genome, which indicates the sequencing of other field-related genetic resources such as wheat pathogens. We performed the quality check of sequenced samples through gene expression data. The library size of the sequenced wheat lines was observed to be variable. However, the logCPM normalisation plot (Figure 1B) and gene expression level distributions both showed that the wheat lines had similar expression patterns and variation ranges suggesting that the sequenced line and sequencing data were comparable and suitable for downstream transcriptome analysis.

To know the effect of different biological and ecological factors of wheat field on wheat leaf transcriptomes, we performed the PCA analysis of wheat lines through TMM normalized expression counts for the wheat genome (Figure 1C). The first PCA component accounted for 71.01% of the total variability, which in this case corresponds to the reference genome specific variance. The second PCA component accounted for 12.76% of the overall variance which highlighted how the difference between the wheat lines might have been caused by other field factors. For a better understanding of the variation among the transcriptome of wheat lines, the MDS plot was used to check variations among samples (Figure 1D). The MDS plot showed that the transcriptome of control lines were very close to susceptible lines transcriptome. In order to identify the candidate genes of wheat among the resistant and susceptible ones, the samples from the control lines were removed from further downstream analysis. Moreover, susceptible and control lines transcriptome resemblance were also supported by field scoring. Although it was clear from the PCA and MDS plots that wheat lines had

a varying expression in the field, it is challenging to make direct comparisons of field factors for transcriptomic variation among wheat lines.

Reference-based transcriptome analysis has been considered more effective than a de novo approach when the reference genome of organism is available [37]. However, very few studies have compared the two strategies [38,39] to identify differentially expressed genes, and no similar comparison has been found for field diseases. Thus, it is more important and interesting for field disease studies to determine whether the de novo assembly can detect the same genes, isoforms and the molecular responses as a reference genome-based analysis, and what else it was captured in transcriptome. In the present transcriptome analysis of wheat, we compared both strategies. The trinity de novo assembly has approximately twice the number of contigs than the reference assembly, which may be due to allelic variation among wheat genotypes, the lack of strand-specific information of genotypes or the sequencing of other unknown eukaryotic field environmental materials. Several genomes and transcriptome assembly studies have used BUSCO for evaluating the quality of assemblies. BUSCO detected the presence of 1347 extremely conserved core eukaryotic genes (CEGs) and their coverage in transcriptome assemblies for the evaluation of the completeness of the assembly. The BUSCO analysis showed that both assemblies were very close to complete in terms of gene content, together with the identified fragments of core-genes. Overall, both assemblies captured high percentages of ultra-conserved core eukaryotic genes (reference: 99.1%, *de novo*: 93.0%). However, the number of complete and single copy BUSCO were found more in de novo assemblies (Table 2). In the transcriptome analysis of wheat lines, the expression abundance and differential expression were different in the de novo and the reference-based analyses. However, there is a larger number of differentially expressed transcripts observed in the susceptible lines compared to the resistant lines (Figures 2 and 3) in both assemblies. De novo and reference-based analysis indicated that the genes expressed in the resistant and susceptible lines might play key roles in their differential resistance abilities. This might be due to the presence of several external factors associated with susceptible lines in the field.

To understand the global trend of genomic variation among wheat lines exposed to different biological and ecological factors in the field, we identified and analysed DEGs from both resistant and susceptible lines in our study through generated reference-based and de novo transcriptome assemblies. The number of identified DEGs in the reference and de novo assembled transcriptome was very low in comparison to the total assembled transcripts, 141 and 8680, respectively, in the resistance and susceptible groups. Gene expression is the primary source to know concerning genetic variation among a population generated during a plant breeding process, and gene expression difference in the population could contribute and represent some of the observed differences in plant phenotype such as susceptibility to diseases. However, a low number of differentially expressed genes might indicate the least genomic variation among resistance and susceptible lines [40] and directly indicate the expression of key genes involved in disease resistance. To identify the potential genes involved in resistance, we investigated upregulated DEGs generated by both transcriptome assemblies. A BLAST similarity search was performed against wheat CDS sequences with the value 1e-10 to determine the commonly expressed genes and unique genes expressed between assemblies. A large proportion of transcript sequences failed to find homologous sequences in the wheat genome. With the used criteria, 126,669 and 107,651 transcript sequences were annotated in de novo and reference assemblies, respectively, in the similarity search of wheat CDS sequences (Table S3 and S4). A large proportion of transcripts sequences generated from wheat lines were failed to find homologous sequences in wheat genome from both the assemblies. Huge number of un-annotating transcripts might be due to inclusion of lot of non-coding RNA content which are missing in wheat gene model or mixing of reads from other eukaryotic organisms.

Approximately 41% of sequences were found to be common between the de novo and reference assemblies, and approximately 29% of sequences were unique in assemblies which might indicate the difference generated by the used assembly approaches, tools and the inclusion of genomic material of other organisms in the wheat transcriptome. Furthermore, both methods identified many similar candidate genes putatively involved in resistant and susceptible genotypes. Thus, this also

demonstrated the potential of the de novo method for the capturing of key genes even in the absence of a reference genome which can be useful for researchers working with field diseases with orphan species. In order to access expression difference in a wheat genotype, a list of upregulated differentially expressed genes was generated by both assemblies (Table 3).Differentially expressed genes from reference and de novo assemblies were annotated by the wheat genome and Uniprot database through BLAST similarity search. It is clearly visible in Table 3 that most of the differentially expressed transcripts related to the disease were identified through de novo transcriptome analysis approach. Cysteine-rich receptor-like protein kinases (CRKs) are most highly differentially expressed in resistant lines among all potential candidates. CRKs are transmembrane, and are involved in a wide range of receptor-like protein kinase-dependent signalling networks including pathogen detection [41]. Seven different isoforms of CRKs; six as CRK6 and one as CRK7 (Table 3) were expressed among resistant lines on different genome locations. The Level of expression varies with respect to the genomic location. However, all the CRKs possess a secretory signal. CRKs have shown a significant role in disease resistance such as NPR1 and NH1 mediated immunity in *Arabidopsis thaliana* and *Oryza sativa subsp. Japonica* against bacterial blight pathogen *Xanthomomas oryzae pv. oryzae*. Absence or low CRKs expression can make plants more susceptible to infection [42]. Diacylglycerol kinase 5 (DAG5) is another gene among other significant DEGs. DAG5 phosphorylates diacylglycerol (DAG) to produce a signalling molecule of phosphatidic acid as a second messenger. Diacylglycerol kinases are key signalling enzymes which are involved in phosphorylating diacylglycerol (DAG) to yield phosphatidic acid (PA). The biosynthesis of PA plays a crucial role in a eukaryotic metabolic and signalling process such as the glycerolipid metabolic process, intracellular signal transduction source, lipid phosphorylation, or the protein kinase C-activating G protein-coupled receptor signalling pathway. Phosphatidic acid is highly required for plant development, abiotic stress and pathogen attack. The presence of diacylglycerol kinase 5 (DAG5) among highly significant DEGs may be an indicator of high resistance among resistant lines [43].

**Table 3.** List of upregulated differentially expressed genes (*p*-value < 0.05) in de novo and reference assemblies that can be considered as potential candidate genes involved in resistance. The table is sorted according to *p*-values. FC: fold change. IWGSC: International Wheat Genome Sequencing Consortium

| Reference/De Novo Transcript | log2FC | Annotation | UniprotIds | IWGSC Chromosome Location | Signal |
|---|---|---|---|---|---|
| TRINITY_DN51078_c2_g4_i2 | 805.981 | Cysteine-rich receptor-like protein kinase | CRK6_ORYSJ | 2D:189356990:189359834:1 | Y |
| TRINITY_DN46223_c0_g1_i25 | 98.303 | Cysteine-rich receptor-like protein kinase | CRK6_ORYSJ | 5A:546234176:546238163:−1 | Y |
| TRINITY_DN46223_c0_g1_i31 | 82.836 | Cysteine-rich receptor-like protein kinase | CRK6_ORYSJ | 5A:546238637:546241834:1 | Y |
| TRINITY_DN46223_c0_g1_i17 | 52.416 | Cysteine-rich receptor-like protein kinase | CRK6_ORYSJ | 5A:546238637:546241834:1 | Y |
| TRINITY_DN38661_c1_g1_i1 | 38.481 | Cysteine-rich receptor-like protein kinase | CRK6_ORYSJ | 2B:245784381:245787768:1 | Y |
| TRINITY_DN45544_c0_g1_i24 | 16.282 | Cysteine-rich receptor-like protein kinase | CRK6_ORYSJ | 2D:20432763:20439635:1 | Y |
| TRINITY_DN37687_c0_g2_i5 | 11.964 | Cysteine-rich receptor-like protein kinase | CRK7_ARATH | 5A:546306021:546309203:−1 | Y |
| TRINITY_DN49928_c0_g1_i18 | 11.545 | Diacylglycerol kinase 5 OS = *Arabidopsis thaliana* | DGK5_ARATH | 2A:748963953:748967651:−1 | N |

| | | | | | |
|---|---|---|---|---|---|
| TRINITY_DN35364_c0_g1_i3 | 2.971 | Disease resistance protein RGA2 OS = *Solanum* | RGA2_SOL BU | 1D:455752931:455755683:1 | N |
| TRINITY_DN44057_c1_g5_i1 | 7.574 | Disease resistance protein RPP8 OS = *Arabidopsis* | RPP8_ARAT H | 6B:662028638:662032091:−1 | N |
| TRINITY_DN39548_c1_g3_i2 | 11.392 | Leaf rust 10 disease-resistance locus receptor-like | LRL28_ARA TH | 3B:29721933:29735756:1 | Y |
| TRINITY_DN35553_c1_g1_i1 | 5.456 | Leaf rust 10 disease-resistance locus receptor-like | LRL28_ARA TH | 6D:2047133:2050237:−1 | Y |
| TRINITY_DN38577_c0_g1_i1 | 12.655 | LRR receptor-like serine/threonine-protein | FLS2_ARAT H | 1D:466386615:466388390:−1 | Y |
| TRINITY_DN38835_c1_g4_i13 | 4.145 | LRR receptor-like serine/threonine-protein | FLS2_ORYSJ | 6D:380568621:380572398:1 | Y |
| TraesCS2B02G608600.1 | 59.366 | Probable LRR receptor-like serine/threonine-protein | Y3475_ARA TH | 2B:788840706:788842030:−1 | N |
| TRINITY_DN41813_c2_g5_i3 | 28.015 | Probable LRR receptor-like serine/threonine-protein | Y3475_ARA TH | 2D:646488160:646491914:1 | N |
| TRINITY_DN51069_c0_g2_i8 | 2.348 | Protein TIFY 6b OS = *Oryza sativa subsp.* | TIF6B_ORY SJ | 5B:369635031:369638011:−1 | N |
| TRINITY_DN35330_c0_g3_i8 | 3.364 | Putative disease resistance protein RGA4 | RGA4_SOL BU | Un:47532810:47545586:1 | N |
| TRINITY_DN51664_c0_g1_i11 | 3.269 | Putative disease resistance protein RGA4 | RGA4_SOL BU | Un:95706705:95715329:1 | N |
| TRINITY_DN35364_c0_g4_i1 | 3.209 | Putative disease resistance protein RGA4 | RGA4_SOL BU | Un:234394428:234397967:1 | N |
| TRINITY_DN51887_c1_g1_i1 | 4.127 | Putative disease resistance RPP13-like | R13L2_ARA TH | 7D:11663563:11672717:1 | N |
| TRINITY_DN52456_c0_g2_i15 | 8.426 | Rust resistance kinase Lr10 OS = *Triticum* | LRK10_WH EAT | 1A:9359231:9363721:1 | N |
| TraesCS6B02G091700.1 | 4.309 | S-(+)-linalool synthase, chloroplastic OS = *Oryza* | LINS_ORYS J | 6B:67408283:67411671:−1 | N |
| TRINITY_DN51301_c1_g1_i10 | 11.432 | Vesicle-associated protein 1-1 OS = *Arabidopsis* | VAP11_AR ATH | 7B:717630687:717634406:−1 | N |
| TRINITY_DN46935_c4_g2_i7 | 5.042 | Vesicle-associated protein 1-1 OS = *Arabidopsis* | VAP11_AR ATH | 7B:717785488:717796197:1 | N |
| TRINITY_DN52084_c1_g1_i8 | 19.697 | Wall-associated receptor kinase 1 OS = *Arabidopsis* | WAK1_ARA TH | 5A:464158592:464170977:−1 | Y |
| TRINITY_DN50322_c1_g1_i11 | 5.429 | Wall-associated receptor kinase 3 OS = *Arabidopsis* | WAK3_ARA TH | 2B:657850507:657854763:−1 | Y |
| TRINITY_DN39541_c5_g2_i2 | 11.016 | Wall-associated receptor kinase 5 OS = *Arabidopsis* | WAK5_ARA TH | 6D:467856066:467860898:1 | Y |

| TRINITY_DN38381_c2_g3_i6 | 5.209 | Wall-associated receptor kinase 5 OS = *Arabidopsis* | WAK5_ARA TH | 6B:713458733:713462914:1 | Y |
|---|---|---|---|---|---|
| TRINITY_DN52648_c2_g2_i1 | 7.132 | Wall-associated receptor kinase-like 9 | WAKLH_A RATH | 6D:467856066:467860898:1 | Y |
| TRINITY_DN51423_c2_g1_i4 | 12.889 | Receptor-like cytoplasmic kinase 185 OS = *Oryza* | RK185_ORY SJ | 2B:104817628:104821546:−1 | N |
| TRINITY_DN50690_c0_g4_i1 | 5.46 | G-type lectin S-receptor-like serine/threonine-protein | Y1130_ARA TH | 2D:642197592:642202812:1 | N |
| TRINITY_DN35935_c1_g1_i3 | 5.033 | G-type lectin S-receptor-like serine/threonine-protein | CE101_ARA TH | 5B:690469413:690490270:−1 | N |

Plant disease resistance in genes encodes two main classes of nucleotide-binding site leucine-rich repeat (NBS-LRR) proteins; TIR-domain-containing (TNL) and CC-domain-containing (CNL). TNLs and CNLs regulate plant resistance through different downstream pathways by inducing a series of defence responses, such as the activation of an oxidative burst, mitogen-associated protein kinase cascade, the induction of pathogenesis-related genes, and the hypersensitive response. In our study, four RGA isoforms; one for RGA2 and three for RGA4, were found among significant DEGs. All four RGAs were found at different genomic locations (Table 3), but all isoforms were having a relatively low but consistent gene expression. RGA isoforms are having an RX-like_CC motif (IPR038005), NB-ARC, LRR domains and belong to the CNL family. The consistent expression of the CNL type of RGAs indicates the "non-race specific" disease resistance and also the activated alternative non-race specific resistance induced by disease resistance proteins RPP8 and RPP13 which indicate a high presence of non-host-specific pathogens in the growing field [44]. LRR receptor-like serine/threonine-protein kinases such as FLS2, Y3475, leaf rust and rust resistance kinases determine the specific perception of pathogen-associated molecular patterns and initiate the innate immune MAP kinase signalling for enhanced resistance against pathogens. The expression of receptor-like cytoplasmic kinase (RLCK185) is clear evidence of the innate immunity triggered by fungal chitin signalling pathways through MAP kinases. Chitin recognition by CERK1 receptor triggers the MAP kinase (MPK3 and MPK6) cascade to search for a host protein that can interact with effector proteins and participates in the activation of defence genes during response to microbial peptidoglycans and chitin [45]. A plasma membrane-associated G-type lectin S-receptor-like serine/threonine-protein kinase (CES101) was also expressed among the significantly expressed DEGs and its involvement was suggested in innate immune response, protein phosphorylation and response to fungus which might have role in the recognition of the lectin-associated molecular pattern of fungus [46]. All RGAs were expressed in the resistant genotypes. Perhaps, the functionality of these RGAs was suppressed by pathogens or lost molecular function due to several genomic modifications in the susceptible genotype. The presence of the RGA4 isoform under an unclassified genome needs to be re-investigated in order to be used as a marker for plant breeding purposes [47]. The expression of protein-like TIFY 6b, known as a repressor of jasmonate responses, is suggesting that plants may be reducing the developmental and metabolic processes for better resource utilization under a biotic stress such as salt tolerance, dehydration and wounding [48]. In general, susceptible cultivars were shown to have a higher gene expression level than the resistant cultivars at the global transcriptome level, which might be due to the higher production and resources allocation against infection and diseases (Figure 6).

The mechanism of cell wall communication to the cytoplasm is not very well known for plant resistance. Wall-associated kinases (WAKs) have the potential to provide clues for the cell wall and the cytoplasm crosstalk. It has been found that WAKs were expressed at organ junctions of shoot, root and leaf in response to wall disturbances, and the expression of an antisense WAK gene in leaves reduced the WAK protein levels and exhibited a loss of cell expansion. The presence of WAKs on the

cell wall is providing the evidence that the receptor-like kinase may have a significant role in the control of cell expansion, morphogenesis and development [49]. In our study, five transcripts were found to be related to wall-associated receptor kinases viz. WAK1, WAK3, WAK5 and WAK9. WAK1 and WAK3 were found on chromosomes 5A and 2B on reverse strands, respectively. The higher expression of WAK1 might show the reduced cell expansion during infection and disease. WAK5 and WAK9 were found on chromosomes 6B and 6D in the forward strand, respectively, which might indicate their involvement in the control of cell expansion, morphogenesis and development under normal circumstances.

The differential gene expression data analysis of wheat lines has increased our understanding of plant defence mechanism in the field and shown evidence for a pathogen-specific and broad-spectrum disease resistance mechanism. Metagenomics analysis of the RNA libraries of samples has also provided evidence for a pathogen-specific disease resistance mechanism along with broad-spectrum disease resistance (Figure 7). In metagenomics analysis, *P. striiformis* was found as the most dominant pathogen among susceptible wheat lines which indicates the susceptibility of wheat lines for this pathogen. The significant differential expression of genes involved in non-race-specific disease resistance among resistant wheat lines provides the evidence for the activation of PAMP-triggered innate immunity (PTI). The presence of other fungal pathogens such as *B. sorokiniana*, *F. culmorum* and *Z. tritici* were supporting the cause of the high activation of innate immunity in the field.

## 5. Conclusions

In the present study, transcriptome analysis was conducted between a resistant and susceptible wheat genotype with a different level of resistance in the field, and several differentially expressed genes were identified through reference and de novo transcriptome assembly. In the transcriptome analysis, the key challenge was the handling of cross-species eukaryotic molecular content of field samples, especially during field infection. In our comparative transcriptome study, the de novo approach was found to be more explorative than the reference assembly process due to high dependency on the reference genome, and the gene expression of similar genes from cross eukaryotic species like fungus has a high chance of influencing the gene expression quantification process due to short reads. In this study, many genes related to plant defence were upregulated in resistant wheat lines in the field. A significant number of genes involved in non-race specific resistance were overrepresented in resistant lines, which might be a reason for the good resistance ability of resistant wheat lines. Expression ofcysteine-rich receptor-like protein kinases, CNL type kinases and wall-associated kinases (WAKs) are suggesting their role in determining broad-spectrum plant defence in the field. GO enrichment and pathway analysis further confirmed that PTI triggered the innate immunity-related genes which were overrepresented in the wheat resistant lines. Potential candidate genes found in this study might provide a basis for future functional host–pathogen genomics research for field wheat. Molecular techniques such as RNA interference can be further used to understand the role of these genes in the field plant resistance.

**Supplementary Materials:** The following are available online at www.mdpi.com/2073-4395/10/12/1888/s1, Table S1: Wheat lines, disease scoring, read count per line, high quality reads and mapping percentage; Table S2: Number of differentially expressed genes among lines with respect to reference and de novo transcriptome; Table S3: Reference transcriptome annotation for wheat genome; Table S4: De novo transcriptome annotation for wheat genome; Table S5: GO enrichment analysis of DEGs; Table S6: Pathway enrichment analysis of DEGs.

**Author Contributions:** S.K.K.: performed the data analysis and wrote the first draft of manuscripts; R.R.V. and F.O.: were responsible for experimental design, implementation, sampling and RNA extractions and manuscript writing; N.P.: helped the manuscript preparation and writing; T.H.: conducted the field trials; A.C. conceived the study, planned the experiment, and secured the funding for this research project. All authors have read and agreed to the published version of the manuscript.

**Funding:** This project was funded by the Lantmännen Research Foundation (2016F010), the Crafoord Foundation (20161025), SLU Grogrund (SLU.ltv.2019.1.1.1-623) and the Royal Physiographic Society of Lund.

**Acknowledgments:** We would like to thank Salvatore Caruso for assistance with the laboratory work. PlantLink, Plant Breeding Platform at SLU, and NIAB, India, are acknowledged for the bioinformatics support.

**Conflicts of Interest:** Authors have declared that they have no competing interest.

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
