# Peer review of "Differential Gene Expression Analysis of Wheat Breeding Lines Reveal Molecular Insights in Yellow Rust Resistance under Field Conditions"

_agronomy, doi:10.3390/agronomy10121888_

Round 1
Reviewer 1 Report
The authors carried out a field experiment and sampled 20 wheat lines for yellow rust resistance using RNA-sequencing. The study is highly interesting and carrying the experiment in field conditions provides new knowledge to the field. However, there were several point which require more clarification. Some of the samples (controls) were mentioned in the text only in the discussion, some of the methods were not explained, and based on the description it is hard to tell whether the data analysis was done in a correct manner. Comparison of de novo assembly vs reference is interesting but not done properly; it is left open whether the new contigs in de novo come from phylloplane microbiome or just result from missing gene annotations in the wheat genome.
Carrying out metagenomic analysis of the transcriptome is a good idea but needs to be followed through properly. After the assembly, the authors should identify the taxonomic assignment for each transcript before carrying out eg comparisons to wheat reference transcriptome.
The analysis of differential expressions should rely more on the GO and KEGG enrichment results, otherwise there's a danger of cherry-picking.
More specifically, I would like to point out the following remarks:
L85 what was the fragment size in the library?
L91. Why didn’t you use the genome-guided Trinity?
L92. How did you assemble the transcriptomes, pooled or separately for each genotype?
L93: Usually after Trinity people do clustering of the gene models because the assembled transcriptome contains allelic variants and partial gene fragments.
L100 For future reference, FPKM is not recommended, TPMs or similar is much better normalisation, see eg https://rnajournal.cshlp.org/content/early/2020/04/13/rna.074922.120.full.pdf
However, (L101) DESeq2 takes in unnormalised counts so FPKMs shouldn’t be affect these
L105: Change Uniport -> Uniprot. This is systematic in the whole MS.
L 106: Name the statistical test for GO enrichment. P-value adjustment for multiple testing should be done, if it wasn’t.
110-113 Metagenomics: this is testing only for the previously identified pathogens and is therefore not proper metagenomics. You should run BLAST against whole NCBI and get taxonomic assignments. The results can be analysed for example with MEGAN6.
L117: But the data was paired end according to M&M?
L120-124: I understand that you want to put all interpretations to discussion but instead of listing things it would be good to include at least some summarizing sentence, eg. based on these results how do the two assemblies compare? You do that in the following pages anyway.
Figure 1: In the M&M you talk about 9 resistant and 11 susceptible lines but here you have also 5 control lines. Where do they come from? c) d) Is this using reference or de novo assembly?
L139: If you’re just aligning against transcriptome, Kallisto or Salmon are nowadays the state-of-the art tools.
L145: “DEGs were blast against” BLAST is an abbreviation, not a verb.
Figure 3: What are the cyan and red coordinations bars in the heatmaps? The two heat maps are now the same size which doesn’t give a proper idea of the differences. Either try scaling the rows to same size or then say on the legend that heatmap A is based on 141 genes and an B 8680 genes (if that’s the case). What was the clustering algorithm?
L177-179: You could have removed the contamination/fungal expression from the wheat diff exp analysis already in the beginning. How many diff exp transcripts do you have left after removal? How many of the original 511,926 isotigs were coming from fungi?
L180-185: These are top categories in GO hierarchy, it’s no wonder you get high percentages. Better to focus on statistically significant enrichments.
L191: REVIGO was not mentioned in M&M and there’s no reference for it.
Figure 5: Add more information to the legend: A) What are the axes “Semantic space X/Y”? How does the color reflect the p-value? The color key is not readable. There’s only one value “-50”. What is blue color and what is red?
B)What is Richness factor? “reflects enriched DGEs” is too obscure.
Figure 6. I would be worried if there would be such a strong correlation between differential gene expression and genomic locus (The innermost track). In all cases the expression increases the further you go along the chromosome. Is this cumulative expression? It also looks like the read depth in resistant lines was much lower than in susceptible lines. Did you test that this doesn’t cause any bias into your analyses?
L217: For proper metagenomic analysis you should run eg MEGAN and get the accurate taxonomic assignments of the non-wheat transcripts.
L219: P. striiformis in italics.
L231-236: If you used de novo assemblies then the phylloplane microbiome can have a significant effect in PCA
L237-240: Why does the circos plot in fig 6 give contrasting message (see my comment above)?
L241-242: Now you mention TMM normalisation when you spoke of FPKM earlier in the text and M&M.
L247: MDS doesn’t do clustering, it’s a nonlinear PCA which projects the data into two dimensions.
L248-250: You need to introduce control lines in the text. Describe in M&M how you got them etc.
L261: “field environmental DNA” - this was RNAseq.
L267-268: “number of complete and single copy BUSCO were found more in de novo assemblies”. Wheat is a hexaploid species so you would expect three copies per Busco gene. This argues that reference-based RNAseq is much better.
L275-300: Comparison of de novo vs. Reference-based: For a proper comparison, you should first identify which transcripts are not coming from wheat. After that filtering, align the remaining transcripts against wheat genome using a splicing-aware aligner and see which ones overlap with gene models. The ones without overlap could be non-coding RNA or non-annotated genes. If you find genes which are not annotated then you can argue that de novo is a better approach.
L300- : Discussion on differential expression: you should start looking at the results top-down, first the GO enrichments, where do they point? Then you look into the significant enrichments and the gene families involved. Otherwise the results could have been produced with cherry-picking as well.
L304-311: Perhaps expand CRK part to include more recent results. CRKs are involved in many processes (https://doi.org/10.1371/journal.pgen.1005373) and one of them is pathogen detection (http://www.plantcell.org/content/32/4/1063)
L320-322: Actually the R genes are split into four classes: https://doi.org/10.1016/j.cell.2019.07.038
L343-344: Pseudogenization would be the most unlikely process for these genes. Their expression could be suppressed by the pathogen or then the host plant doesn’t induce their expression.
Reviewer 2 Report
The manuscript 'Differential gene expression analysis of wheat breeding lines reveal molecular insights in yellow rust resistance under field condition'. by Sandeep et al. compared de novo and reference methods analysis transcriptome data to identify the resistance genes in bi-parental population lines. The sufficient data was present and the comparison between two assembling methods is interesting. However, I can not accept the manuscript for publication in current form.
Major issue,
1) There is no data validation of identified candidate genes in parental lines or in the bulked susceptible and resistance lines.
2) How about the sequence variations between the susceptible and resistance pools of lines? if you could include the SNP-index value of each chromosomes, between susceptible and resistance pools, you will find the candidate delta SNP index distribution peaks (areas) and 'G' value distribution area. Then scanned these candidate loci with genome sequence. Therefore, match the scanned genes from genome sequence to your listed candidate genes (DEG) from transcriptomes. I think from this strategy, you could minimize the number of candidate genes. Also, it will be an effective way to identify the candidate genes.
Minor issue (just several examples):
1) all the term indicate gene should be italic.
2) In introduction, you should include the information about how many Yr genes have been cloned? and the progress of recently studies.
3) L72-73: The crossed parents (combination) is not clear. Please divide them by ',' or 'space'.
4) L77: '-180' or '-80', I guess '-80'
5): L168: Arabidopsis thaliana and oryza sativa should be italic.
6) L283-284:This sentence could be strongly supported or not? just from low number of DEGs?
7) L309: Arabidopsis thaliana and oryza sativa should be italic.
.
.
.
Round 2
Reviewer 1 Report
L93: Usually after Trinity people do clustering of the gene models because the assembled transcriptome contains allelic variants and partial gene fragments.
Response: All the generated transcripts were clustered through Cd-hit software at 100% sequence identity.
REFEREE COMMENT: Please add this information to the text. Tell also the number of isotigs before and after clustering.
10-113 Metagenomics: this is testing only for the previously identified pathogens and is therefore not proper metagenomics. You should run BLAST against whole NCBI and get taxonomic assignments. The results can be analysed for example with MEGAN6.
Response: We are agree with reviewer for full metagenomics analysis. But in this study, the exploration of yellow rust and other known wheat pathogens was our focus to infer susceptibility and resistance of genotype with respect to pathogenic precedence in field. Therefore, by purpose, we limited our analysis to very common known wheat pathogens only.
REFEREE COMMENT: At least this should be stated in the text. For example: “To explore field pathogenic factor association with gene expression of wheat lines, a metagenomic analysis was performed among resistant and susceptible line samples by mapping the non-aligning reads to known wheat pathogens with sequenced genome information.” - or similar.
L117: But the data was paired end according to M&M? Response: Sorry, we were unable to connect comment with manuscript text in line 117. But yes, paired end read sequencing was done for the study.
REFEREE COMMENT: This point referred to this sentence on line 117: “A total number of 307 million reads of average 126 bps were generated from the 20 lines and 207 million high-quality reads were used in the construction of transcriptome assembly.” This sounds line you did single-end 126 bp sequencing, not eg. 2x125bp paired end. Please correct this in the text.
Figure 1: In the M&M you talk about 9 resistant and 11 susceptible lines but here you have also 5 control lines. Where do they come from? c) d) Is this using reference or de novo assembly?
Response: In this study, total 25 wheat genotype [9 resistant, 11 susceptible and 5 as control] were sequenced and scored for yellow rust. Resistance and susceptible genotype were decided on the basis of yellow rust score. We observed that susceptible and control lines were have very similar yellow rust score during scoring process. In PCA and MDS plot, similar observation were found that control lines were very close to susceptible lines. All the sequenced genotype was pooled to make assembly. But to explore resistance and susceptible lines molecular difference, control lines were excluded from differential gene expression analysis.
REFEREE COMMENT: All of this needs to be explained in the M&M. Now you only say you had 109 lines with three plants each. How do you end up with the 25 lines?
Figure 3: What are the cyan and red coordinations bars in the heatmaps? The two heat maps are now the same size which doesn’t give a proper idea of the differences. Either try scaling the rows to same size or then say on the legend that heatmap A is based on 141 genes and a B 8680 genes (if that’s the case). What was the clustering algorithm?
Response: Red and cyan colour group represents the resistant and susceptible lines respectively. Pheatmap, R package was used for heatmap generation. Euclidean clustering distance and hierarchical clustering algorithm was used.
REFEREE COMMENT: The edit to the legend was made but the clustering algorithm is not explained in the M&M, please add it there. Also tell which algorithm you used (Ward, single-linkage, complete-linkage…?)
L177-179: You could have removed the contamination/fungal expression from the wheat diff exp analysis already in the beginning. How many diff exp transcripts do you have left after removal? How many of the original 511,926 isotigs were coming from fungi?
Response: Initially, reads were mapped over known wheat pathogens. Later, mapped reads were removed for transcriptome assembly. All the constructed transcript were annotated against wheat genome and wheat annotated transcripts form differential gene expression analysis were considered further downstream analysis. Wheat annotation for transcripts: Reference (107651) and de novo (126669) can be found in supplementary file.
REFEREE COMMENT: Please then explain the mapping stragy in the M&M, perhaps by adding to this sentence: “To facilitate more in-depth comparison of the two different genotypes, reference based transcriptome assembly was generated through the alignment of all clean reads from each line to the reference wheat genome by using HiSat2 and Stingtie software “ Also say that you had 511,926-126,669 = 385257 de novo isotigs which didn’t map to wheat and you then explored that using targeted metagenome analysis.
Figure 5: Add more information to the legend: A) what are the axes “Semantic space X/Y”? How does the color reflect the p-value? The color key is not readable. There’s only one value “- 50”. What is blue color and what is red?
Response: X and Y coordinates were derived by applying multidimensional scaling to a matrix of the GO terms' semantic similarities which reflect their closeness in the GO graph structure. Colour represent significance of GO terms and size indicates the frequency of the GO term in the underlying Gene Ontology Annotation (GOA) database. Most significant GO term represent by blue colours whereas least significant by orange colour. Higher the frequency of GO term in GOA database, larger the circle size such as larger circle in blue are most frequent and significant enriched GO term.
REFEREE COMMENT: You explain this now twice in the text, once in the main body and second time in figure legend. Remove this from the text: “In figure 5a, colour represent significance of GO terms and size indicates the frequency of the GO term in the underlying gene ontology annotation database such as larger circle in blue are most significant enriched term. X and Y coordinates were based on multidimensional scaling of matrix of the GO terms' semantic similarities whereas bubbles closeness on the plot were reflecting their closeness in the GO graph structure. “..”Enriched pathway among differentially expressed genes were provided as scatter diagram with degree of enrichment by the rich factor, p-adjust, and number of genes enriched in a pathway (Figure 5b). The richness factor represents the ratio of the quantity of genes belonging to the pathway among differentially expressed genes to the total number of genes belonging to the pathway among all annotated genes. Lower the p-value, higher the significance of metabolic pathways, whereas the size of bubbles represents the number of enriched” and make sure this information is conveyed in the figure legend. The main text is for results, not describing what we see in the figures.
In figure 5a, you should still describe what the colours notify. The color key has only one value -50 at green color. What does the red color correspond to, higher or lower log10(p) ?
L248-250: You need to introduce control lines in the text. Describe in M&M how you got them etc.
Response: These controls are just checks which are often used in the field trials as a way to see if there is a disease pressure. The only purpose of having the controls was to confirm that there is a disease pressure. Further analysis of controls was not done as it is a different genetic material than what is being studied in this work. Moreover, infection score of control line were very close to susceptible lines and similar outcome also have seen in PCA plot. Therefore, Control lines were not used in differential gene analysis.
REFEREE COMMENT: OK, this is fine. Just explain this in the M&M because you will be talking about the control lines in the text (See also my comment above).
L 280-282: Trinity de novo assembly has approximately twice the number of contigs than reference assembly may be due to the lack of strand specific information of genotypes and sequencing of other unknown eukaryotic field environmental
REFEREE COMMENT: Most likely this is due to allelic variation between genotypes, you’re just assembling heterozygous alleles as their own gene models. It is also possible that wheat annotation is missing some gene models. Please include also these possibilities.
L267-268: “number of complete and single copy BUSCO were found more in de novo assemblies”. Wheat is a hexaploid species so you would expect three copies per Busco gene. This argues that reference-based RNAseq is much better.
Response: Relatively large number of complete duplicate genes in both assemblies is likely due to the polyploidy of the genome whereas complete single copy genes were more in de novo genes which might be because of fragmented transcripts. All the reference and de novo assembled transcripts were annotated against wheat genome through BLAST similarity search at evalue 1e-10 and their expression and significance were extracted from expression analysis A large proportion of transcripts are common in both the assemblies [Reference (107651) and de novo (126669)] (Supplementary file). But when we looked to individual gene level, it has several mismatches and gap which might be generated during assembly process or mixing of reads from other eukaryotic organisms. We also rephrased the sentences in manuscripts
REFEREE COMMENT: Based on text you did the alignment against the CDS, not the genome. If you did alignment against genome, then you should use a splice-aware aligner such as Exonerate or PASA instead of BLAST. I’m now assuming you did alignment against CDS. In that case revise the sentence: “A large proportion of transcripts sequences failed to find homologous sequences in wheat genome. “, instead of gene say “predicted transcriptome” or similar.
If you don’t find a match, one obvious possibility is that the gene model is missing or, since you looked at total RNA, that you have a lot of non-coding RNA which is not included in wheat (protein coding) gene models. Please comment this in the manuscript.
REFEREE COMMENT: I appreciate the changes in the CRK discussion but please revise the grammar.

Reviewer 2 Report
I very much appreciate of the authors could address most of my comments properly. However, for the major concerns I threw out in the first round review, the authors emphasis to do/conduct these following research later. From my view of point, most of the scientific journals can not accept the transcriptome analysis without data validation. Even now, a lot of journals can not accept accept the RNA seq data with simple qPCR validation. Recently, they will request functional analysis of the candidates.
I can understand the authors used two different methods to highlight the candidate genes. The methods and comparisons are interesting. However, without experimental validation, I will still can not accept it for publication in my opinion.